# Evaluating Large Language Models on Controlled Generation Tasks

**Jiao Sun**[1*] **Yufei Tian**[2*] **Wangchunshu Zhou**[3*] **Nan Xu**[1*]
**Qian Hu**[4] **Rahul Gupta**[4] **John Wieting**[5] **Nanyun Peng**[2] **Xuezhe Ma**[1]

[1]University of Southern California  [2]University of California, Los Angeles
[3] ETH Zurich  [4] Amazon  [5] Google DeepMind

{jiaosun,nanx,xuezhema}@usc.edu {yufeit,violetpeng}@cs.ucla.edu
wangchunshu.zhou@inf.ethz.ch {huqia, gupra}@amazon.com
jwieting@google.com

## Abstract

While recent studies have looked into the abilities of large language models in various benchmark tasks, few studies have looked into the controllability of large language models on generation tasks. We present a systematic and extensive analysis of the controllability of large language models on ten benchmarks, including a new simple yet challenging numerical planning benchmark with different granularities. After comparing large language models against state-of-the-start finetuned smaller models, we present a spectrum showing when large language models fall behind, are comparable, or exceed the ability of smaller models. We conclude that **large language models struggle at meeting fine-grained hard constraints**.

## 1 Introduction

Text generation models should generate texts that meet controllable constraints as humans wish (Zhang et al., 2022). For example, one can avoid the blandness caused by repetitive patterns by controlling the syntax of generated sentences (Iyyer et al., 2018; Qian et al., 2019). In a customized dialogue system, one should be able to control the persona of the utterance (Smith et al., 2020). Previous works either finetune generation models such as BART (Lewis et al., 2019) on specific tasks for better controllability (e.g., controlled paraphrase generation (Sun et al., 2021)) or design constrained decoding strategies (e.g., look-back decoding strategy by Xu et al. (2023a)) for controlled generation.

Large Language Models (LLMs) have recently shown great potential in various generation tasks. For example, Jiao et al. (2023a) shows that `ChatGPT` with GPT-4 as an engine achieves commercial-level machine translation quality. Laskar et al. (2023) find that annotators prefer summaries generated from `ChatGPT` over state-of-the-art summarization models. However,

| Task | Control | Benchmark | Evaluation |
|------|---------|-----------|------------|
| constrained content generation | sentiment, topic, keyword | Amazon Review CommonGen M2D2 | off-the-shelf model, ppl |
| story generation | prefix | ROC writing prompts | repetition, diversity, coherence |
| rationale generation | correct answer | CoS-E ECQA | increased accuracy |
| numerical planning (NEW) | prefix & number of words & end word | NPB | MSE, success rate |
| paraphrase generation | semantic & syntax | ParaNMT QQPPoS | lexical overlapping, syntax match |

good ➡ poor

Figure 1: We test large language models on five controlled generation tasks with various control factors using automatic evaluation methods. We show a spectrum of abilities of large language models on such tasks and conclude that large language models struggle at fine-grained hard constraints such as numerical planning.

few works investigate the controllability of large language models. Towards this end, we aim to study and understand the controllability of large language models to answer the question: *Are large language models better than finetuned smaller models at controllability on generation tasks?*.

The main contribution of this work is to conduct a comprehensive analysis of LLM's controllability on five tasks and ten generation benchmarks, including controlled story generation, controlled free-form generation with sentiment and topics, controlled paraphrase generation, and controlled rationale generation as in Figure 1. We further design a new simple yet challenging benchmark named Numerical Planning Benchmark (NPB), where the task is to satisfy numerical constraints from four granularities (word-, syllable-, sentence- and paragraph-level) and under different content controls (e.g., prefix and ending). For evaluation, we use automatic metrics, which are imperfect yet convenient and reproducible.[1]

---

*The first four authors contribute equally.

[1] https://github.com/sunjiao123sun/

After an in-depth examination, we categorize LLM's controllability on a spectrum: from lagging behind and being on par with to surpassing smaller finetuned models. Our findings indicate that large language models have difficulties adhering to specific hard constraints, such as numerical planning.

We first introduce the numerical planning task and the associated evaluation as this is a new, intuitively simple, yet challenging task (§2). For the rest, we rank them by the task difficulty indicated in Figure 1 from easy to hard: constrained content generation (§3), story generation (§4), rationale generation (§5) and paraphrase generation (§6).

## 2   Numerical Planning

*Can LLMs count from two to ten?*

**Task Description.**   We introduce the Numerical Planning Benchmark (NPB) as an intuitive task that tests the *basic numerical planning ability* of LLMs. The high-level task descriptions can be found in Table 1. We are inspired by real-world scenarios such as creative writing. For example, writers may wish to generate sentences or poems with a specific structure, such as a fixed number of words or syllables in each line, aiming to adhere to particular forms (*e.g.,* sonnets, where each line contains exactly 10 or 11 syllables (Tian and Peng, 2022)). Meanwhile, humans may also want full control over the start and end of each line for rhetorical purposes such as alliteration and rhyming. Inductively, we formulate our numerical planning benchmark from four different granularities: generating a piece of text that contains a predefined number of *words, syllables, sentences, or paragraphs* given a plausible pair of prefix (start) and suffix (ending) as constraints. The prefix is given to LLMs such that they are only queried to generate the continuations.

**Evaluation Metrics.**   We use success rate (SR) and mean squared error (MSE) as automatic evaluation metrics. As our control is two-fold, we separately calculate the success rates of 1) generating the continuation with the correct counts and 2) generating the continuation with the proper ending. We also calculate the MSE between our input numbers and output numbers.

**Evaluate with LLMs.**   We evaluate `ChatGPT` and `Alpaca-7b` on our NPB benchmark in zero-shot and few-shot settings. Each request used to query the LLMs corresponds to a real case in the datasets

| Granularity | Task Illustration |
|---|---|
| Word/Syllable | Generate a sentence using exactly 5 words/syllables. |
| | Complete sentence "This is a story" using exactly 5 words/syllables. |
| | Complete sentence "This is a story" using exactly 5 words/syllables, including the last word as "town". |
| Sentence | Generate a paragraph with 5 sentences, ... |
| Paragraph | Generate an article with 5 paragraphs, ... |

Table 1: Task illustration for the Numerical Planning Benchmark. We test LLMs' numerical planning ability under various constraints (word counting and end word) and granularities (word, syllable, sentence, and paragraph). Due to space limitations, we only show the full constraints under the word granularity here.

of Romance Books and Reddit Short Stories.[2] For word-level planning tasks (word and syllable count), we randomly select sentences from the above datasets. Then, we select the last word in each sentence as the suffix. Depending on how many additional words we query the LLMs to generate, we select the first few words in each sentence as the prefix (if we simply ask LLMs to generate freely without a prefix, the outputs lack diversity). Our prompt is written as *Complete a sentence that starts with {prefix} using exactly {N} additional words (including the last word {last word}). The sentence must end with the word {last word}. Sentence: {prefix}*, and LLMs will continue. In the few-shot setting, we provide the task description and three examples. For each example, we also provide explanations to help LLMs better understand our task. For example,

*##Prefix: This is a story about a young girl's*
*##Last word: town*
*##N: 5*
*##Output: This is a story about a young girl's redemption in a small town.*
*##Explanation: We generated "redemption in a small town". It contains exactly 5 words and ends with the last word 'town'.*

We query the LLMs to generate outputs from $N = 2$ to $N = 10$ words. Each number $N$ has 100 evaluation samples. For paragraph-level tasks, the prefix and suffix are the first and last sentences in the corresponding paragraphs. For all experi-

---

[2]huggingface.co/datasets/AlekseyKorshuk/romance-books, www.kaggle.com/datasets/trevordu/reddit-short-stories

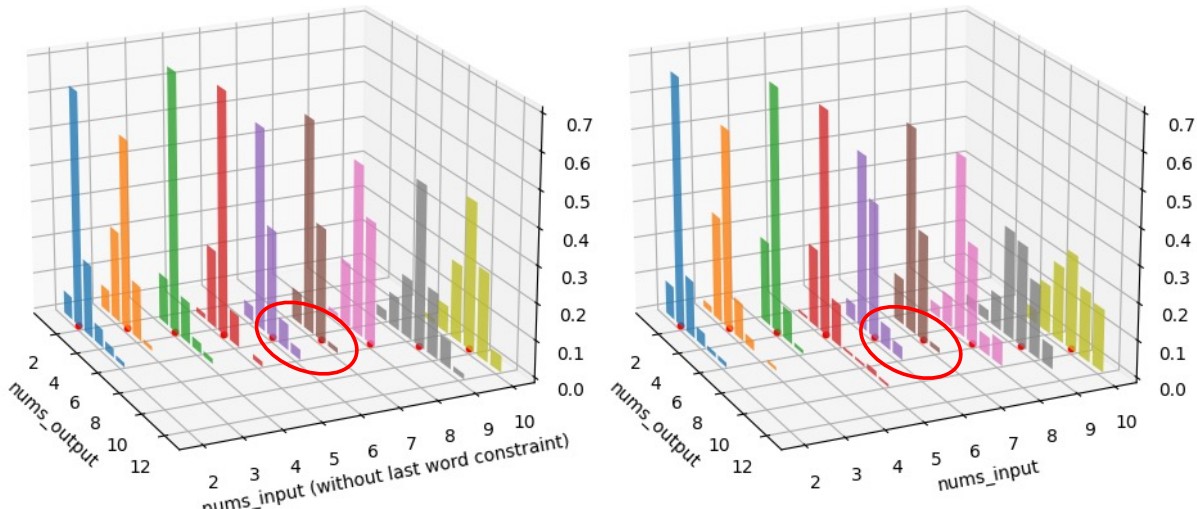

Figure 2: Histogram visualization in the distribution (frequency, z-axis) of input numbers (x-axis) and output numbers (y-axis) for word count planning. Left: querying ChatGPT to generate a continuation of a given prefix with $N$ words. Right: querying ChatGPT to generate a continuation with $N$ words of a given prefix that ends with a given word. Small red dots • mark those bars where output numbers equal input numbers. These bars represent the fine-grained success rates. For either case, there is a significant drop when the input number reaches six.

| Model | SR - count | SR - last word | SR - both | MSE - count |
|---|---|---|---|---|
| GPT-2 (fine-tuned) | 0.64 | 0.86 | 0.60 | 1.62 |
| Alpaca-7b$_{zs}$ | 0.17 | 0.31 | 0.09 | 9.19 |
| Alpaca-7b$_{ICL}$ | 0.14 | 0.34 | 0.07 | 9.76 |
| Vicuna$_{zs}$ | 0.08 | 0.09 | 0.03 | 27.68 |
| Vicuna$_{ICL}$ | 0.13 | 0.30 | 0.04 | 13.43 |
| Falcon$_{zs}$ | 0.13 | 0.42 | 0.08 | 11.60 |
| Falcon-7b$_{ICL}$ | 0.11 | 0.34 | 0.03 | 13.72 |
| ChatGPT | **0.41** | 0.74 | **0.36** | **3.64** |
| ChatGPT$_{ICL}$ | 0.37 | **0.78** | 0.34 | 4.95 |

Table 2: Success rates for the word count planning task. Surprisingly, few-shot in-context learning (ICL) underperforms zero-shot (zs) on numerical planning.

ments, our decoding strategy is top $p$ ($p = 0.95$) sampling with temperature $T = 0.3$ unless otherwise specified.

**Result.** We report the model performance of LLMs and a fine-tuned GPT-2-large model on the task of word count planning in Table 2. Due to space limitations, we compile the results of the remaining tasks in Appendix A. First, it is clear LLMs are poor at numerical planning, although it is an extremely simple task for humans. Given its extremely poor performance, we consider Alpaca incapable of doing numerical planning. Secondly, LLMs learn to incorporate literal constraints, such as the last word, via few-shot in-context learning. Interestingly, *few-shot in-context learning deteriorates the performance of numerical planning*.

Upon further inspection, we find that LLMs try to mimic the style or features (such as length) in the in-context examples and are, therefore, more likely to generate outputs with the wrong word counts once the input number $N$ cannot be found in the examples. Our results resonate with Yin et al. (2023); Kung and Peng (2023); Sinha et al. (2023) that LMs do not truly understand task definitions via in-context learning.

Figure 2 is a fine-grained visualization of the input and output numbers distribution by zero-shot ChatGPT. Specifically, we compare LLMs' numerical planning abilities with (e.g., *complete sentence with "redemption in a small town" using exactly 5 words, including the last word as "happy"*) and without additional suffix constraint (e.g., *complete sentence with "redemption in a small town" using exactly 5 words*). LLMs can generate more freely without suffix constraints to meet the numerical constraint. However, ChatGPT doesn't always translate to a higher success rate. We find out that only when $N$ is small (i.e., 2 and 3), ChatGPT achieves a higher success rate if explicitly told the last word of the target sentence.

Finally, we would like to point out a few behaviors. First, although the general trend is that LLMs' numerical planning ability drops as $N$ increases, $N = 3$ is a clear exception (performs worse) among various experiments we repeated. Second, by checking the failure cases, we find that

`ChatGPT` *always generates shorter continuations* than required. Moreover, we see a sudden drop in model performances (from above $\sim$0.6 to $\sim$0.4) when the input number $N$ increases from 5 to 6. We encourage future research to investigate these behaviors.

## 3 Content-Controlled Generation

**Task Description.** We consider three types of content constraints: topic, sentiment, and keyword. The detailed task definitions and dataset can be found in Appendix B.

**Evaluation Metrics.** We use the success rate as the evaluation metric to measure how well LLMs can follow the content constraints. Specifically, we use GPT-3.5 (Ouyang et al., 2022) based topic/sentiment classifiers with in-context learning using five examples per category to evaluate whether the generated texts belong to the specified topic or sentiment class. We consider an LLM to succeed in one example if the predicted class of the generated text is identical to the input constraint. For a keyword-constrained generation, we use the keyword coverage metric that measures the percentage of input keywords included in generated texts.

**Evaluate with LLMs.** For the content constrained generation with LLMs, we follow Zhou et al. (2023) and use natural language instructions to prompt LLMs. Specifically, we use a prompt template of *"Write a sentence about {topic name}"* for topic-constrained generation, *"Write an Amazon review with {level number} star about a random thing. The number of stars ranges from one to five. One star is the most negative, and five stars are the most positive"* for sentiment constraints, and *"Write a sentence using the following keywords: {keywords}"* for keyword constraints.

In addition to zero-shot evaluation, we also evaluate LLMs in the in-context learning setting by appending the following demonstration template: *"Below are some examples for the task: Input: {input 1}, Output: {output 1}; Input: {input 2}, Output: {output 2} ... "*. We use 5 in-context examples per class following the practice in Zhou et al. (2023).

We compare various LLMs including `ChatGPT`, LLaMA, Alpaca, Vicuna, and Falcon in our experiments. We also report the results of Diffusion-LM (Li et al., 2022b) based on BERT-large (Devlin

| Model | Topic | Sentiment | Keyword |
|---|---|---|---|
| Diffusion-LM | 68.9 | 83.7 | 93.2 |
| GPT-2 (1.5B, fine-tuned) | 63.4 | 76.5 | 88.9 |
| T5 (3B, fine-tuned) | 67.3 | 83.9 | 94.8 |
| LLaMA-7B$_{zs}$ | 45.3 | 58.4 | 83.5 |
| LLaMA-7B$_{ICL}$ | 63.5 | 85.1 | 93.0 |
| Alpaca-7B$_{zs}$ | 58.9 | 78.4 | 91.2 |
| Alpaca-7B$_{ICL}$ | 65.2 | 86.9 | 94.8 |
| Vicuna-7B$_{zs}$ | 61.0 | 80.5 | 91.6 |
| Vicuna-7B$_{ICL}$ | 65.8 | 87.4 | 94.3 |
| Falcon-7B$_{zs}$ | 61.9 | 81.0 | 92.1 |
| Falcon-7B$_{ICL}$ | 66.0 | 87.7 | 94.2 |
| ChatGPT$_{zs}$ | 66.4 | 84.5 | 97.3 |
| ChatGPT$_{ICL}$ | **88.4** | **90.3** | **98.1** |

Table 3: Results on content-constrained text generation.

et al., 2019) and task-specific classifiers as a competitive non-LLM baseline

**Results.** The results are shown in Table 3. We find that Alpaca significantly outperforms LLaMA in the zero-shot setting. This is intuitive since natural language instruction of constraints resembles instruction tuning data. However, this performance gap is significantly reduced when in-context learning is used. We think this is because the role of instruction tuning is mainly to adapt an LLM to human-friendly prompt formats instead of increasing the LLM's capability. We also find that `ChatGPT` achieves competitive performance without in-context learning and outperforms Diffusion-LM, a competitive supervised baseline, by a large margin. Moreover, the performance of `ChatGPT` can be further improved by adding in-context examples to the prompt. This suggests that LLMs' ability to follow content constraints expressed in natural language depends on three confounding factors: instruction tuning or supervised fine-tuning, in-context learning, and model capacity.

## 4 Story Generation

**Task Description.** Given the beginning text of a story, open-ended story generation aims to decode texts that are coherent with previous topics, and informative without undesired repetitions (Su et al., 2022; Su and Xu, 2022; Xu et al., 2023b). Despite the impressive success on generating fluent and accurate sentences for low-entropy tasks such as summarization or translation, large-scale language models (LLMs) still suffer from serious degeneration problems, such as undesired repetitions (Holtzman et al., 2020; Su et al., 2022) and

| LM | Method | rep-2↓ | rep-3↓ | rep-4↓ | diversity↑ | coherence↑ |
|---|---|---|---|---|---|---|
| | | | **ROC** | | | |
| | Human | 1.74 | 0.32 | 0.04 | 0.97 | 0.48 |
| GPT-2-XL | Nucleus | 1.80 | 0.35 | 0.12 | 0.97 | 0.33 |
| | Typical | 2.06 | 0.4 | 0.16 | 0.97 | 0.33 |
| | $\eta$-sampling | **0** | **0** | **0** | **1.00** | 0.34 |
| | SimCTG | 3.10 | 0.46 | 0.23 | 0.96 | 0.32 |
| | Look-back | 7.24 | 0.92 | 0.14 | 0.92 | 0.47 |
| LLM | Vicuna | 2.36 | 0.45 | 0.15 | 0.97 | 0.60 |
| | Falcon | 2.52 | 1.87 | 1.86 | 0.94 | **0.69** |
| | ChatGPT | 1.18 | 0.10 | 0.02 | 0.98 | 0.52 |
| | | | **Writing Promts** | | | |
| | Human | 15.61 | 3.78 | 1.24 | 0.80 | 0.31 |
| GPT-2-XL | Nucleus | 5.40 | 2.41 | 1.72 | 0.91 | 0.34 |
| | Typical | 3.60 | 1.51 | 1.10 | 0.94 | 0.30 |
| | $\eta$-sampling | 6.17 | 2.88 | 2.16 | 0.89 | 0.35 |
| | SimCTG | **2.84** | **0.36** | **0.19** | **0.97** | 0.31 |
| | Look-back | 7.94 | 1.25 | 0.33 | 0.91 | 0.52 |
| LLM | Vicuna | 8.27 | 2.59 | 1.14 | 0.88 | 0.49 |
| | Falcon | 11.20 | 7.79 | 6.94 | 0.76 | **0.53** |
| | ChatGPT | 5.99 | 1.15 | 0.35 | 0.92 | 0.52 |

Table 4: Performance of different decoding strategies and LLMs for open-ended story generation. Vicuna stands for Vicuna-7B, Falcon for Falcon-7B-Instruct.

unnatural topic drifts (Li et al., 2022a), under open-ended settings.

**Datasets.** We evaluate different generation methods on two popular benchmark story datasets: ROCStories and Writing Prompts. ROCStories (ROC) (Mostafazadeh et al., 2016) is a corpus comprising commonsense stories written by crowd-sourced workers within 5 short sentences. Given the first sentence as a prefix, generation methods are required to produce four continuing sentences. Writing Prompts (WP) is a challenging task for inspiring continuations with abstract, high-level story prompts submitted by online users and continuations by others on Reddit (Fan et al., 2018). Following prior literature (Xu et al., 2023b), we utilize the first 32 tokens as the prefix and ask for continuation with 256 tokens. Since we prompt different language models or decoding algorithms without extra fine-tuning, we directly sample 1,000 development and 1,000 testing instances from both ROC and WP.

**Baselines.** We evaluate the pre-trained LLM, GPT-2-XL (Radford et al., 2019), with both search (SimCTG (Su et al., 2022) and Look-back (Xu et al., 2023b)) and sampling decoding methods (Nucleus sampling (Holtzman et al., 2020), Typical decoding (Meister et al., 2022) and $\eta$-sampling (Hewitt et al., 2022)).

**Evaluation Metrics.** Following open-ended story generation literature (Su et al., 2022; Li et al., 2022a; Xu et al., 2023b), we adopt the following automatic metrics to evaluate generation quality: 1) *rep-n* to measure sequence-level repetition according to the portion of duplicate n-grams (Welleck et al., 2019); 2) *diversity* to assess the overall model repetition by considering *rep-n* at different n-gram levels; 3) *coherence* measured as the cosine similarity between prefix and continuation embeddings represented by SimCSE (Gao et al., 2021). We do not report MAUVE (Pillutla et al., 2021) score due to the concern that MAUVE may not accurately reflect human preferences considering contradicted results between MAUVE and human evaluations observed in prior work (Su and Xu, 2022).

**Evaluate with LLMs.** Chatbots that fine-tune LLMs on instructions are also evaluated: Vicuna-7B (Chiang et al., 2023), Falcon-7B-Instruct (Almazrouei et al., 2023) and ChatGPT. [3] We prepend the following instruction before the story prefix as prompt: 1) ROC: "Please continue writing this story within 4 very short sentences: <prefix>", 2) WP: "Please continue writing this story within 256 words: <prefix>"[4].

**Results.** As shown in Table 4, both Vicuna-7B and ChatGPT are able to continue writing more fluent and coherent stories on both ROC and WP compared with other decoding methods based on GPT2-XL. Falcon-7B-Instruct obtains consistently lower diversity than other baselines, while ChatGPT achieves more robust performance in terms of diversity and coherence on both datasets.

## 5 Rationale Generation

**Task Description.** Free-form rationales are known to aid model interpretability by providing additional world knowledge or commonsense reasoning steps (Kim, 2015; Lipton, 2018; Alvarez-Melis and Jaakkola, 2018). Wei et al. (2022) show that rationales can improve large language models' ability to solve complex reasoning tasks. Extractive rationales in question-answering tasks are based on the input passage to extract related information to answer the question. Conversely, free-form rationales in the question-answering tasks are open-

---

[3] https://chat.openai.com/
[4] We adopt generation parameters for different LLMs suggested from their respective documents or APIs. We leave evaluation on more configurations in our repository: https://github.com/sunjiao123sun/llm-controlgen.

| | Leakage | Non-Leakage |
|---|---|---|
| $I{\rightarrow}O$ | 0.87 | |
| $I+R_{CoS\text{-}E}{\rightarrow}O$ | 0.92 | |
| $I+R_{ECQA}{\rightarrow}O$ | **0.99** | |
| **Model** | **Leakage** | **Non-Leakage** |
| $I+R_{Alpaca\text{-}7B}{\rightarrow}O$ | 0.91 | 0.86 |
| $I+R_{LLaMA\text{-}7B}{\rightarrow}O$ | 0.87 | 0.79 |
| $I+R_{Vicuna\text{-}7B}{\rightarrow}O$ | 0.95 | 0.74 |
| $I+R_{Falcon\text{-}7B}{\rightarrow}O$ | 0.83 | 0.65 |
| $I+R_{ChatGPT}{\rightarrow}O$ | **0.98** | **0.93** |

Table 5: Rationales generated by ChatGPT are on par with best-crowdsourced rationales ECQA with FlanT5-XXL (Chung et al., 2022b) as the backbone model. Ruling out leakage results in at least 5% accuracy drop.

ended and condition on purely the question and options. (Sun et al., 2022) studies how different the quality of rationales would impact rationales' utilities in terms of improving the model performance and claims that crowdsourced rationales are superior to generated rationales. Sun et al. (2022) finetunes T5-base for both rationale generation and question answering. With the power of LLMs, we want to revisit the problem and see whether the utility of generated rationales conditioned on the question and options has been improved.

**Evaluation.** We follow previous works and use the performance gap before and after adding rationales in the input to measure the utility of rationales, written as acc(I+R→O) - acc(I→O), where I stands for question and options as input, R stands for rationales, and O stands for one of the options as output. For the backbone model for question answering, we use flanT5-XXL (Chung et al., 2022a) instead of T5-base as it can handle longer sequences and is better at reasoning.

Sun et al. (2022) shows that two factors are mainly affecting the utility of rationales. One is *leakage*, which means that the correct answer is explicitly written in the rationales, and one can choose the correct answer among all the options by rationales without knowing the questions. The other is *background knowledge*, which is the additional background knowledge or reasoning step that can help answer the question.

**Datasets.** CoS-E (Rajani et al., 2019) and ECQA (Aggarwal et al., 2021) are the most popular free-form rationale datasets through crowdsourcing. ECQA builds on CoS-E and improves the quality of the CoS-E dataset from various aspects, including completeness, comprehensiveness, coherence, etc.

They share the same sets of questions and options. Based on the findings from Sun et al. (2022), both CoS-E and ECQA tend to leak the correct answer in the rationale, while ECQA rationales contain the background necessary to answer the questions. We conduct our analysis on question-answer pairs from the test set. Based on the evaluation acc(I+R→O) - acc(I→O), since we are evaluating on the same set of question-answer pairs, acc(I→O) is always the same. Therefore, we only compare acc(I+R→O) with different LLMs.

**Evaluate with LLMs.** We prompt LLMs to provide background knowledge that can help answer the question and control whether to leak the correct options in rationales. We use ChatGPT as the example for illustration:

- *Leakage.* We have ChatGPT take the role of *A teacher who is trying to explain to students the rationale behind choosing the correct option for a multiple-choice question.* Then prompt it with *Question: {question} Options: {concatenated options} Explain the rationale behind choosing the correct option "{correct answer}"*.

- *Non-leakage.* The role of ChatGPT becomes *A teacher who is trying to explain to students the rationale behind a multiple-choice question. However, you do not want to leak the correct answer directly.* and prompt it with *Question: {question} Options: {concatenated options} Explain the rationale behind choosing the correct answer. Do not mention the correct answer "{correct answer}" explicitly*.

We highlight the difference between the two modes with underline. When prompting LLaMA and Alpaca, we remove the role description and only use the prompts. Through analysis, we aim to answer two questions: 1) Are LLM-generated rationales on par with crowdsourced rationales? 2) How much would leakage impact the utility of rationales?

**Result.** Compared to T5, FlanT5 has better reasoning abilities (Chung et al., 2022b) and is more capable of understanding instructions. Therefore, we use FlanT5 instead of using T5 as the backbone model for question answering, which can theoretically examine the utility of rationales better ruling out the incapability of models. Simply given the question and the option strings, Table 5 shows that FlanT5-XXL has an accuracy of 0.87

(while T5 in (Sun et al., 2022) scores 0.57 under the same setting). We then show the performance with crowdsourced rationales from both ECQA and CoS-E. With crowdsourced rationales from ECQA, the model almost solved the task and reached a performance of 0.99. With CoS-E rationales, the accuracy is 0.92. Our finding echoes with Sun et al. (2022) that ECQA rationales are better quality.

We then evaluate the utility of LLM-generated rationales under both the *Leakage* and *Non-leakage* scenarios. As the majority of crowdsourced rationales contain leakage (Sun et al., 2022), we consider it fair to compare LLM-generated rationales under the *Leakage* scenarios against crowdsourced rationales. We have two major findings:

- `ChatGPT` generated rationales are on par with ECQA rationales from crowdsourcing.

- We quantify the influence of leakage in measuring the utility of rationales: whether or not having leakage in rationales could result in an accuracy difference of at least 5%.

## 6   Controlled Paraphrase Generation

**Task Description.**   Syntactically-controlled paraphrase generation can benefit a wide range of NLP applications such as dialogue generation (Gao et al., 2020), improving the robustness of models (Huang and Chang, 2021) or metrics (Aggarwal et al., 2022), and diversifying other generation tasks such as diverse question generation. Syntactically-controlled paraphrase generation is challenging because it requires satisfying two folds of control signals: semantic preservation and syntactic conformation. By definition of paraphrases, the generation should have exactly the same semantics as the input text. With syntax as part of the input, generated paraphrases should also conform with the indicated syntax. The input syntax can come from a variety of sources.

**Datasets.**   We evaluate on ParaNMT-small (Chen et al., 2019), derived from ParaNMT (Wieting and Gimpel, 2018), and QQP-Pos (Kumar et al., 2020). Our train/dev/test split follows previous works (Kumar et al., 2020; Sun et al., 2021). Each instance is a tuple of {source sentence, exemplar, ground-truth paraphrase}, where the exemplar shares the same syntax with the ground-truth paraphrase.

**Evaluation Metrics.**   We use two sets of evaluation metrics to evaluate the quality of generated paraphrases. We use lexical-overlapping-based scores to evaluate the semantic preservation and tree-edit distances to evaluate the syntactic conformation. For lexical-overlapping-based scores, the higher is better. For tree edit distance, the lower is better, indicating that the newly derived syntax matches more closely with the expected syntax. In this work, we prune the constituency parse trees at a level of 2 and only compare the high-level syntactic structure. TED-R means the tree edit distance between the candidate-generated sentence with the ground-truth paraphrase as the reference. TED-E compares the candidate sentence against the exemplar that only provides the syntax.

**Evaluate with LLMs.**   We provide three ways to prompt for the controlled paraphrase generation:

- *Direct.* We prompt LLMs directly without specifying any constraints. The prompt is written as *Paraphrase {source sentence}. Please only have the paraphrase in the response.*

- *Control.* Under this mode, we use the exemplar sentence for the syntactic control signal. The prompt is written as *Paraphrase "{source sentence}" so that it uses the syntactic structure from "{exemplar}"; please only have the paraphrase in the response.*

We observe that under the *Control* mode, the generated paraphrases would sometimes take the syntactic information from the exemplars and the semantic meaning from exemplar sentences. To solve this, we introduce the third mode *Control with syntax explanation*. We first extract the constituency parse structure from the exemplar sentence using Stanford CoreNLP, prune the parse tree at the height of two (i.e., parse at H2), and then ask `ChatGPT` to generate a natural language explanation of the pruned syntactic parse, which we refer to as *syntax explanation*. The generated syntax explanation will be part of the input.

- *Control with Syntax Explanation.* The prompt is written as *Paraphrase "{source sentence}" so that the sentence has a syntactic structure of "{pruned syntax}". {generated explanation for the syntax.} Please only have the generated paraphrase, not its parse, in the response.*

Table 7 shows examples of generated explanations for constituency parse trees pruned at height

| | | BLEU↑ | METEOR↑ | ROUGE-1↑ | ROUGE-2↑ | ROUGE-L↑ | TED-R↓ (H=2) | TED-E↓ (H=2) |
|---|---|---|---|---|---|---|---|---|
| ParaNMT -Small | Direct | 10.8 | 26.2 | 44.2 | 18.6 | 44.9 | 1.4 | 1.5 |
| | Ctrl | 14.3 | 30.7 | 51.4 | 25.8 | 50.7 | 1.3 | 1.2 |
| | Syntax exp. | 13.6 | 27.3 | 46.4 | 20.2 | 47.0 | 1.4 | 1.4 |
| | 🏆 AESOP | **22.9** | **32.7** | **54.4** | **29.8** | **56.4** | **0.9** | **0.5** |
| QQPPos | Direct | 6.7 | 25.2 | 39.8 | 15.6 | 41.5 | 1.8 | 1.8 |
| | Ctrl | 10.5 | 25.6 | 43.0 | 19.8 | 45.2 | 1.4 | 1.4 |
| | Syntax exp. | 9.0 | 26.5 | 42.8 | 17.8 | 14.2 | 1.8 | 1.8 |
| | 🏆 AESOP | **47.3** | **49.7** | **73.3** | **54.1** | **75.6** | **0.4** | **0.3** |

Table 6: Performance comparison with ground-truth syntactic control for AESOP (Sun et al., 2021) and fine-shot `ChatGPT`. With coarse syntactic control from a shallow height of pruning, AESOP, the state of the finetuned small model, outperforms five-shot `ChatGPT` across **all** semantic preservation (BLUE, ROUGE Scores, and METEOR) and syntactic conformation metrics (TED-R and TED-E at the height of two) by a large margin. ↑ means higher is better, while ↓ means lower is better. By comparing *ctrl* with *syntax explanation*, we show that `ChatGPT` is better at mimicking the syntactic structure from an exemplar than utilizing the syntactic information directly from the syntax.

| Pruned Parse at H=2 | Explanation |
|---|---|
| (ROOT (S (NP ) (VP ))) | This represents a sentence structure with a noun phrase and a verb phrase as its constituents. |
| (ROOT (FRAG (SBAR ) (. ))) | This is a sentence with a fragment that includes a subordinate clause followed by a period. |
| (ROOT (SBARQ (WHADVP ) (SQ ) (. ))) | This sentence structure represents an interrogative sentence with a subord -inate clause before the main clause. |
| (ROOT (SQ (VBP ) (RB ) (NP ) (VP ) (. ))) | This is a parse tree for a sentence containing a main verb and its subject, with a possible adverb and complement structure. |

Table 7: Examples of generated explanations for pruned constituency parse trees by `ChatGPT`.

two by `ChatGPT`. We prompt `ChatGPT` from zero shots to five shots for our experiments, find that `ChatGPT`'s performance peaks with five shots as expected, and compare the performance of five-shot `ChatGPT` with AESOP (Sun et al., 2021). The backbone of AESOP is the BART-base model, a 140m-parameter model finetuned with specialized input and output format tailored for the controlled paraphrase generation task. To the best of our knowledge, AESOP remains the state-of-the-art paraphrase generation model on both ParaNMT-small and QQPPos datasets.

**Result.** Table 6 shows the performance comparison between five-shot `ChatGPT` and AESOP. We show that AESOP surpasses `ChatGPT` across all evaluation metrics for both semantic preservation metrics (lexical-overlapping based metrics including BLEU, ROUGE scores, and METEOR) and

syntactic conformation metrics (TED-R and TED-E at the height of two). In addition, we find that `ChatGPT`'s performance is the best under the setting of *Control*, where we use exemplar sentences for control signals. Compared with the setting *Control with syntax explanation*, Table 6 shows that `ChatGPT` is good at mimicking syntactic structures from sentences instead of directly incorporating the syntactic parses. Besides `ChatGPT`, we also tried Alpaca (Taori et al., 2023) and LLaMA (Touvron et al., 2023) on the controlled paraphrase generation task. However, they repeat input sentences and struggle to generate meaningful content. Therefore, we do not include them here for comparison.

## 7 Related Works

**LLM Evaluation.** While the advancement of more potent large language models drives our work, our focus aligns more with recent studies evaluating LLMs' performance on academic NLP benchmarks. We roughly categorize these studies as either general or specific NLP tasks. For general NLP tasks, Qin et al. (2023) shows that `ChatGPT` performs well on many tasks involving reasoning capabilities but not on sequence tagging. Ahuja et al. (2023) evaluate LLMs on various multilingual NLP tasks. For specific tasks, Jiao et al. (2023b) shows that `ChatGPT` has achieved competitive performance on machine translation. Gao et al. (2023) uses `ChatGPT` for event extraction and shows that it only matches with around a half percent of specialized event extraction models. To the best of the authors' knowledge, we are the first to study the controllability of LLMs and the tasks in our work

have not been previously studied. Instead of having a single conclusion on if LLMs perform well at certain task, we provide a spectrum showcasing how LLMs' abilities vary according to different control granularities.

## 8 Discussion: Why and How

We believe that our work makes a substantial contribution to the field of benchmarking LLMs' controllabiltiy, especially considering the prevalence of LLMs these days. That being said, we do have a few hypotheses to investigate ***why*** LLMs fail at numerical planning and ***how*** we could potentially increase their controllability.

**Tokenization.** On one hand, tokenization indeed makes the task of numerical planning more challenging than without, by separating the generative process (*i.e.,* subword-level generation) and the numerical planning process (*i.e.,* counting complete words). However, we posit that tokenizers not necessarily impact the ability of word planning, as it is a standard practice that a subword starting with a special token will indicate the start of a new word (*e.g.,* "Ġ" in BPE tokenizer,[5] which has been used by many LLMs such as GPT and RoBERTa). Nor are we aware of evidence that the subwords of a tokenizer roughly correspond to units of syllables. For example, Tian et al. (2023) shows that smaller models such as GPT-2-large fine-tuned on syllable-related data can achieve a success rate of close to 90% on the same syllable-planning task. On the other hand, the best performance of ChatGPT is 37%.

**Decoding Methods.** The reported results are based on sampling with a temperature of 0.3. Moreover, we have experiments showing that our conclusion is robust to the change of decoding mechanisms, where we try other decoding methods beyond sampling with $T = 0.3$.

Specifically, we tried 1) greedy decoding, 2) beam search with beam size 8, and 3) sampling with temperature $T = \{0.3, 0.7, 1.0\}$. For the prior two, most of the generated outputs are highly similar, plain, and lack diversity. As for sampling with $T = \{0.3, 0.7, 1.0\}$, the success rate decreases as $T$ increases. We think $T = 0.3$ is a reasonable balance between diversity and quality. We believe that our results convey meaningful signals since each

number $N$ has been averaged over 100 different evaluation samples to reduce noise. However, none of these experiments show that LLMs can do better than fine-tuned GPT-2.

**In-Context Learning.** We try to give more demonstration of NPB in our prompts and we surprisingly found that this does not help once the input number $N$ cannot be found in the examples. Our results resonate with Yin et al. (2023); Kung and Peng (2023) that LLMs do not truly understand task definitions via in-context learning.

**How to Improve.** We encourage future work to explore from two different directions: 1) chain/tree/graph-of-thought reasoning, and 2) bridging LLMs with non-autoregressive generation abilities (e.g., NADO (Meng et al., 2022)). For the first one, one can try both simple chain/tree/graph-of-thought prompting or even pretrained LLMs with chain-of-thought/scratchpad pairs, as it shows promises for mathematical reasoning (Zhou et al., 2022). However, this will not fundamentally solve the planning issue. It is straightforward that auto-regressively generating the next tokens will lead to the problem of models not "looking back" and therefore not adhering to the fine-grained control signals. Therefore, we encourage researchers to also investigate multi-step planning and iterative revisions with LLMs, or, more fundamentally, challenge the autoregressive architecture of LLMs.

## 9 Conclusion

We test the controllability of large language models on five tasks and ten benchmarks, including a numerical planning benchmark that is easy for humans while challenging for LLMs. From there, we draw a spectrum by comparing the performance between LLMs and smaller specialized models. LLMs are able to generate human-level rationales and conform with coarse control signals, such as sentiment, topic and keyword incorporation. However, they struggle at fine-grained hard constraints, such as numerical planning and paraphrase generations. We hope that our work can inspire downstream applications on when to adopt LLMs. For example, we find that LLMs are good at generating rationales, and these automatic rationales could be used to further boost LLMs' performance through chain-of-thought reasoning.

---

[5] https://huggingface.co/learn/nlp-course/chapter6/5?fw=pt#byte-pair-encoding-tokenization

## Acknowledgement

The authors thank anonymous reviewers for their constructive feedback and suggestions that helped improve the draft, especially reviewer rXWW. Jiao and Yufei are supported by Amazon fellowships.

## Limitations

This work is subject to couple of limitations. First, all of our experiments involved heavy prompt engineering effort. Although we have attempted to choose the best performing prompts, there might be room for better prompts which could influence the reported evaluation metrics. Second, automatic evaluations are imperfect. Last, we have not proposed solutions after identifying tasks where LLMs struggle. We leave this for future work.

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

| Model | SR - count | SR - suffix | SR - both | MSE - count |
|---|---|---|---|---|
| syllable planning | | | | |
| ChatGPT | 0.37 | 0.75 | 0.32 | 4.83 |
| ChatGPT ICL | 0.30 | 0.84 | 0.28 | 6.10 |
| Alpaca-7b | 0.15 | 0.33 | 0.07 | 9.44 |
| Alpaca-7b ICL | 0.12 | 0.36 | 0.05 | 10.61 |
| sentence planning | | | | |
| ChatGPT | 0.38 | 0.625 | 0.29 | 1.69 |
| ChatGPT ICL | 0.36 | 0.66 | 0.27 | 2.05 |
| Alpaca-7b | 0.19 | 0.19 | 0.07 | 6.56 |
| Alpaca-7b ICL | 0.17 | 0.26 | 0.10 | 8.04 |
| paragraph planning | | | | |
| ChatGPT | 0.69 | 0.17 | 0. | 3.24 |
| ChatGPT ICL | 0.57 | 0.17 | 0.34 | 4.43 |
| Alpaca-7b | Failed | | | |
| Alpaca-7b ICL | Failed | | | |

Table 8: Success rates for the syllable, sentence, and paragraph count planning tasks. LLMs are best at sentence count planning and worst at syllable count planning.

## A  SPB additional results

We report the additional results of ChatGPT and Alpaca on the SPB benchmark in Table 8. Recall that the suffix for the paragraph planning task is the last sentence. In practice, LLMs are unable to follow instructions and copy the requirement as prompted. Hence, when we compute the success rate for this last task, we check the token overlap between the generated sentence and our requirement, and if more than 2/3 of the tokens overlap, we will consider it as a success.

Taking all four tasks in the SPB benchmark into account, we find out that Alpaca-7b have very little numerical planning ability. ChatGPT on the hother hand is best at sentence count planning, and worst at syllable count planning.

## B  Additional Information of Content Controlled Generation

Controlled content generation refers to the task of controlling the content of generated texts. We consider three types of content constraints:

- *Topic constraint.* It requires the model to generate texts about certain topics. Traditional methods for topic constrained generation either append a special token for different topics (Çağlayan and Karakaya, 2021) or use trained topic classifiers (Qin et al., 2022) to guide the generation process.
- *Sentiment constraint.* Similar to topic constraint, this task requires the model to generate texts of

certain sentiments. The aforementioned methods for topic constrained generation also apply to sentiment constrained generation.

- *Keyword constraint.* Keyword constrained, or lexical constrained text generation requires the model to generate texts that contain certain keywords or tokens. Traditional methods for keyword constrained text generation generally enforce lexical constraints on the outputs by modifying the search space according to the constraints (Anderson et al., 2017; Post and Vilar, 2018; Lu et al., 2021).

**Datasets.** For topic constraints, we use a subset of the topics from the first hierarchy in the M2D2 dataset (Reid et al., 2022) which contains domains such as health, history, society, technology, arts, science, etc. The total number of topics is 10 in our experiments. We use the Amazon Review dataset (Keung et al., 2020) for sentiment constrained text generation. The sentiment is measure by 1 to 5 stars. For lexical constrained text generation, we use the CommonGEN dataset (Lin et al., 2020) which requires the model to generate a sentence using three to five keywords.