# OpenReview forum: "Evaluating Large Language Models on Controlled Generation Tasks"
_EMNLP/2023/Conference — EMNLP 2023 Main_

### Official Review · Reviewer_Hweb · 2023-08-04

**Soundness:** 4

**Excitement:**

3: Ambivalent: It has merits (e.g., it reports state-of-the-art results, the idea is nice), but there are key weaknesses (e.g., it describes incremental work), and it can significantly benefit from another round of revision. However, I won't object to accepting it if my co-reviewers champion it.

**Paper Topic And Main Contributions:**

This paper evaluates LLMs on controlled generation tasks, stepping into five unexplored tasks. Among them, the authors propose a new benchmark called “numerical planning” which is hard for LLMs.

Contribution：
1. Study the controllability of large language models using five unexplored tasks.
2. Propose a new controlled generation benchmark which is hard for LLMs.

**Questions For The Authors:**

1. In the evaluation of Rationale Generation, FlanT5-XXL has an accuracy of 0.87, is this too high to show less differences between models? How to get a better balance between the shortages of backbone model and its reasoning abilities?
2. Till now, which LLM should also be considered in these controlled generation tasks?

**Reasons To Accept:**

1. The evaluation is really detailed and well designed.
2. Figure 1 shows a very good overview of the whole paper, and other tables are quite clear.
3. The writing is good and the structure is nice.

**Reasons To Reject:**

1. The appendix should include more details, such as more examples and presentation of failed examples.

**Reproducibility:**

3: Could reproduce the results with some difficulty. The settings of parameters are underspecified or subjectively determined; the training/evaluation data are not widely available.

**Reviewer Confidence:**

4: Quite sure. I tried to check the important points carefully. It's unlikely, though conceivable, that I missed something that should affect my ratings.

**Typos Grammar Style And Presentation Improvements:**

Line 218, one more period
Maybe adding subtitles in Figure 2 will be better
Check the bold number of “diversity” in the table below of Table 4, should be the biggest one

---

> ### Author Rebuttal · Authors · 2023-08-29
>
> We really appreciate your suggestions and would like to clarify some details as follows:
>
> **Reason to reject: Include qualitative examples**
>
> Good point! We will add some failure examples for challenging tasks, including numerical planning and controlled paraphrase generation, in our final version.
>
> **Question: FlanT5 too strong? Balance the shortages of backbone model and its reasoning abilities**
>
> This is a great point. We deliberately choose a strong backbone model that has strong reasoning capability to make sure that the performance difference comes from the rationale quality instead of the reasoning incompetence of the backbone model. We are confident that FlanT5 is an appropriate discriminator choice for this task because it responds properly to 1) the presence of rationales and 2) the change of rationale quality. From Table 5: Adding rationales helps boost the model performance, suggesting that FlanT5 is not too strong to solve the task directly. Based on [1], the quality of ECQA rationales is better than CoS-E, resulting in a better model performance. Therefore,  we believe FlanT5 strikes the perfect balance between reasoning and model capability for the rationale generation task!
>
> [1] Sun, Jiao, et al. "Investigating the Benefits of Free-form Rationales." EMNLP 2022.
>
>
> **Q2: Recent LLMs to be checked**
>
> Although there are new open-source LLMs coming at a fast pace, they are arguably not exceeding the performance of ChatGPT (https://chat.lmsys.org/?leaderboard). On the other side, as we respond to R1 under Reason 3, R2 under Reason 3, we decide the LLM sets as {ChatGPT,  LLaMA, Alpaca, Falcon, and Vicuna} throughout the paper and conduct complementary experiments to make our study more consistent and comprehensive. Unsurprisingly, we find that both Vicuna and Flacon still underperform ChatGPT. Thus, additional results do not affect our conclusion in the paper. Please find our new results here:
>
> Table 2 for Numerical Planning Benchmark (NPB):
>
> | **Model**	 |  **SR-Count**	 | **SR-Last word**|	**SR-both** |**MSE-count**|
> | -------- | ------- | ------- | ------- | ------- |
> | Vicuna_{zs}	 |  0.08		 |  0.09		|  0.03	|	27.68|
> | Vicuna_{ICL}  |  0.13		 |  0.30		|  0.04	|	13.43|
> | Falcon_{zs}    |  0.13		 |  0.42		|  0.08	|	11.60|
> | Falcon_{ICL}  |  0.11		 |  0.34		|  0.03	|	13.72|
>
> Table 3 for Content-Constrained Text Generation:
>
> |**Model**           | **Topic**  |   **Sentiment**  |     **Keyword** |
> | -------- | ------- | ------- | ------- |
> |Vicuna_{zs}  |	61.0	|           80.5	      |     91.6 |
> |Vicuna_{ICL} | 65.8	|	87.4	      |     94.3 |
> |Falcon_{zs} 	|61.9	|           81.0	      |	92.1 |
> |Vicuna_{ICL} 	| 66.0	|	87.7	      |	94.2 |
>
> Table 5 for Rationale Generation:
> | **Model**              |          **Leakage**      |     **Non-Leakage** |
> | -------- | ------- | ------- |
> | I + R_{Vicuna} |		0.95	        |	 0.74   |
> | I + R_{Falcon} |		0.83	        |	 0.65   |

---

### Official Review · Reviewer_rXWW · 2023-08-04

**Soundness:** 4

**Excitement:**

4: Strong: This paper deepens the understanding of some phenomenon or lowers the barriers to an existing research direction.

**Missing References:**

The finding that ChatGPT mimics the syntactic structure of the sentences in its context is related to the following paper which you might be interested in:

Koustuv Sinha, Jon Gauthier, Aaron Mueller, Kanishka Misra, Keren Fuentes, Roger Levy, and Adina Williams. 2023. Language model acceptability judgements are not always robust to context. In Proceedings of the 61st Annual Meeting of the Association for Computational Linguistics (Volume 1: Long Papers), pages 6043–6063, Toronto, Canada. Association for Computational Linguistics. URL: https://aclanthology.org/2023.acl-long.333/

**Paper Topic And Main Contributions:**

This paper attempts to characterize the controllability of the text generated by LLMs, and in particular answers the question: “Are LLMs better than fine-tuned smaller models at controllability on text generation tasks?”

To this end, the authors design a comprehensive spectrum of tasks that sheds light on the extent to which their text generation mechanisms can be made to adhere to constraints ranging from specific attributes like sentiment, keywords, etc., to precise numerical planning, to the syntax and semantics of the output. This maps to 6 different experiments (11 different tasks), each of which targets a specific controllable constraint along with the evaluation of specific characteristics (e.g., repetition, diversity, lexical overlap, syntactic matches, etc.). As part of the experiments, the authors also design a benchmark called the Numerical Planning Benchmark that adds to existing work on controllable text generation by devising instructional prompts that specifically target the number of words/sentences/paragraphs in the generated output. Specifically for this experiment, the authors find that the tested LMs were generally bad at numerical planning, and that interestingly few-shot in-context learning actually hinders performance on this task. ChatGPT to a small extent does achieve success but this is limited to when the number of units to plan for (words/sentences, etc.) is small, and the last token is provided.

Overall, the findings suggest that LLMs are more likely to be controlled to generate text that obey soft/coarse grained constraints such as sentiment, topic, generation of rationales, etc.,  but struggle on finer-grained constraints such as numerical planning and paraphrasing.

**Questions For The Authors:**

A: According to the primary research question, LLMs are being compared to “finetuned smaller models” -- are these models fine-tuned on the specific task? If yes, then you are not really comparing against any such model in Sections 3,4, and 5.

B: How reasonable is it to expect LLMs to reason about “syllables”? Wouldn’t this depend primarily on the tokenizer more than anything?

C: Did you try other decoding methods for the NPB experiments? I am not sure if results from a single setting can be counted as fully comprehensive.

D: (feel free to not answer this if there is no space): GPT-2 XL gets perfect scores on ROC for a majority of the metrics in the story generation experiments using $\eta$-sampling (as shown in Table 4) -- this seems peculiar and interesting, do you have an idea as to why this might be happening?

**Reasons To Accept:**

- The authors present a spectrum of different controllable text generation scenarios for evaluating LLMs -- this seems like a neat, novel empirical contribution that can shed light on the ways in which LLMs lack in satisfying the constraints presented to them in their inputs.
- Each task involves careful consideration of the design of the experiment as well as the dataset and measures used to shed light on the particular type of controllable generation -- the authors should be commended for this!
- Within the spectrum, the authors further propose a novel dataset called the Numerical Planning Benchmark which targets the ability of LLMs to plan their generation based on precise numerical factors present in the prompt/instruction.

**Reasons To Reject:**

- The research question states that LLMs are being compared to “finetuned smaller models” but there is no such comparison being made in sections 3, 4 (this one uses pre-trained GPT-2), and 5 (all rationales are generated using off the shelf pre-trained LLMs and these are compared against crowdsourced rationale generations) **Note:** *This point of weakness hinges on the answer to Question A of the next section.*
- It is not immediately obvious what determines the position of a task on the spectrum. For instance, the rationale generation performance of LLMs was quite good and should probably be closer to “good” than “poor” in Figure 1.
- The LLMs being compared are quite different across the 6 sections -- for instance sec 2 compares ChatGPT, and Alpaca, while section 3 compares ChatGPT, Alpaca, and LLaMa, while section 4 compares ChatGPT, Vicuna, and Falcon, ..and so on. This seems to defeat the purpose and potential of the full spectrum of tasks which is supposed to collectively demonstrate a holistic comparison between LLMs on controllable text generation. It would have been better if the authors chose a select few LMs and only compared them throughout the tasks (against task-specific smaller fine-tuned LMs, to stay faithful to the main research question).

**Reproducibility:**

4: Could mostly reproduce the results, but there may be some variation because of sample variance or minor variations in their interpretation of the protocol or method.

**Reviewer Confidence:**

4: Quite sure. I tried to check the important points carefully. It's unlikely, though conceivable, that I missed something that should affect my ratings.

**Typos Grammar Style And Presentation Improvements:**

The title seems to be grammatically incorrect, I think you meant “Evaluating” instead of “Evaluate” -- I hope you can change this!!

Line 577: “chain of thoughts” → “chain of thought”

**Important:** I would seriously like to see a high level discussion on how the results and analyses demonstrate the usefulness of the spectrum. Currently, the conclusion section seems to pop out suddenly and unnaturally after the final experiment. I strongly encourage taking this into account for the additional page, if the paper is accepted/made public. I say this because there is genuine value in having a full range of tasks with varying difficulty and amounts of control, and it would be great if the paper does more to shed light on this fact.

---

> ### Author Rebuttal · Authors · 2023-08-29
>
> Thank you for such a detailed and insightful review. We appreciate your insightful suggestions and precious feedback!
>
> **Reason 1 and Q. A: Comparison between LLMs and fine-tuned smaller models**
>
> We compare LLMs with task-specific baselines, where fine-tuned smaller models are included most of the time. For other tasks, we tailor based on the common practice or which performs the best in that task. To make it more rigorous, we will change the phrasing to ``Comparison between LLMs and task-specific models” for our final version.
>
> Content-Controlled Generation (Section 3): For fine-tuned smaller models, we included the performance of Diffusion-LM in the first row of Table 3, which uses fine-tuned classifiers as score functions. We also tested fine-tuned GPT-2 and T5-3B with the results listed below. We observe that LLMs slightly underperform strong fine-tuned smaller models such as T5 in the zero-shot setting, and the performance gap is very small when LLMs are equipped with in-context examples.
> |**Model**           | **Topic**  |   **Sentiment**  |     **Keyword** |
> | -------- | ------- |------- |------- |
> | GPT-2 (1.5B, fine-tuned) | 63.4 | 76.5 | 88.9 |
> | T5 (3B, fine-tuned) | 67.3 | 83.9 | 94.8 |
>
> Story Generation (Section 4): We do not include fine-tuned models since in the open-ended setting, recent research work focuses on decoding strategies with zero training [1] because 1) decoding alone could achieve **similar good performance** as fine-tuned models [2] and 2) decoding is more efficient and feasible for improving generation of LLMs while fine-tuning methods are computationally prohibitive, which greatly limits their practical applicability [3].
> Rationale Generation (Section 5): Rationale generation is indeed an exception itself, where we directly compare LLMs with crowdsourced rationales due to two reasons: 1) finetuned T5 perform poorly at rationale generation task [4], and 2) the crowdsourced rationales do not necessarily mean high quality and are comparable to the quality of ChatGPT generation. Such contrast supports our claim well that LLM is very capable of rationale generation, approaching human performance. We will clarify the uniqueness of this task in our final version.
>
> [1] Xu, Nan, et al. "Look-back Decoding for Open-Ended Text Generation." arXiv preprint arXiv:2305.13477 (2023).
> [2] Su, Yixuan, and Nigel Collier. "Contrastive search is what you need for neural text generation." TMLR 2023.
> [3] Li, Xiang Lisa, et al. "Contrastive decoding: Open-ended text generation as optimization." ACL 2023.
> [4] Sun, Jiao, et al. "Investigating the Benefits of Free-form Rationales." EMNLP 2022.
>
> **Reason 2: Spectrum of LLM abilities**
>
> Awesome catch, we will change the notation for rationale generation in Figure 1 from red to green.
>
> **Reason 3: Selection of LLMs for comparison keeps changing**
>
> Great point! As we respond to R1 under Reason 3, we decide on the LLM sets as {ChatGPT,  LLaMA, Alpaca, Falcon and Vicuna} throughout the paper and conduct complementary experiments to make our study more consistent and comprehensive. Unsurprisingly, we find that both Vicuna and Flacon still underperform ChatGPT. Thus, additional results do not affect our conclusion in the paper. For paraphrase generation, ChatGPT performs the best and all other models do not generate reasonable paraphrases, we therefore keep our original results untouched. We will merge the new results with the current results in our final version.
>
> Table 2 for Numerical Planning Benchmark (NPB):
>
> | **Model**	 |  **SR-Count**	 | **SR-Last word**|	**SR-both** |**MSE-count**|
> | -------- | ------- | -------- | ------- |------- |
> | Vicuna_{zs}	 |  0.08		 |  0.09		|  0.03	|	27.68|
> | Vicuna_{ICL}  |  0.13		 |  0.30		|  0.04	|	13.43|
> | Falcon_{zs}    |  0.13		 |  0.42		|  0.08	|	11.60|
> | Falcon_{ICL}  |  0.11		 |  0.34		|  0.03	|	13.72|
>
> Table 3 for Content-Constrained Text Generation:
>
> |**Model**           | **Topic**  |   **Sentiment**  |     **Keyword** |
> | -------- | ------- |------- |------- |
> |Vicuna_{zs}  |	61.0	|           80.5	      |     91.6 |
> |Vicuna_{ICL} | 65.8	|	87.4	      |     94.3 |
> |Falcon_{zs} 	|61.9	|           81.0	      |	92.1 |
> |Vicuna_{ICL} 	| 66.0	|	87.7	      |	94.2 |
>
> Table 5 for Rationale Generation:
> | **Model**              |          **Leakage**      |    **Non-Leakage** |
> | -------- | ------- |------- |
> | I + R_{Vicuna} |		0.95	        |	 0.74   |
> | I + R_{Falcon} |		0.83	        |	 0.65   |
>
>
> **Q. B: Whether reasonability of reasoning about “syllables” primarily depends on tokenizer**
>
> As far as we know, the ability of syllable planning does not necessarily depend on tokenizers. Nor are we aware of evidence that the subwords of a tokenizer roughly correspond to units of syllables. For example, [1] shows that smaller models such as gpt-2-large fine-tuned on syllable-related data can achieve a success rate of close to 90% on the same syllable-planning task. On the other hand, the best performance of ChatGPT is 37%.
>
> [1] Tian, et al. Unsupervised Melody-to-Lyric Generation. https://aclanthology.org/2023.acl-long.513.pdf
>
> **Q. C: Other decoding methods for the NPB experiments**
>
> We did try different decoding algorithms on the NPB benchmark in the early experimental stage but decided to proceed with the most reasonable setting: sampling with T=0.3.
>
> Specifically, we tried 1) greedy decoding, 2) beam search with beam size 8, and 3) sampling with temperature T={0.3, 0.7, 1.0}. For the prior two, most of the generated outputs are highly similar, plain, and lack of diversity. As for sampling with T={0.3, 0.7, 1.0}, the success rate decreases as T increases. We think T=0.3 is a reasonable balance between diversity and quality. We believe that our results convey meaningful signals since each number N has been averaged over 100 different evaluation samples to reduce noise.
>
> **Q. D: GPT-2 XL gets perfect scores on ROC in Table 4**
>
> First of all, story generation is relatively easier compared with other controlled generation tasks as demonstrated in Table 1. Secondly, ROC dataset is not as challenging as Writing Prompts as shown in Table 4 because 1) ROC is an easier corpus comprising commonsense stories while Writing Prompts dataset is far more challenging for inspiring continuations with abstract, high-level story prompts (from lines 254 to 265); 2) GPT-2 released in 2019 scraped web pages as training data, where ROC released in 2016 could be included while samples from Writing Prompts were not covered based on our rough check. We will add prior discussion in our work.
>
> **Missing Reference and typos**
>
> Great catches! We will fix all these in our final version and do another round of careful proofreading.

---

### Official Review · Reviewer_iLd9 · 2023-08-05

**Soundness:** 4

**Excitement:**

3: Ambivalent: It has merits (e.g., it reports state-of-the-art results, the idea is nice), but there are key weaknesses (e.g., it describes incremental work), and it can significantly benefit from another round of revision. However, I won't object to accepting it if my co-reviewers champion it.

**Paper Topic And Main Contributions:**

This paper study the task of controlled generation, evaluating the performance of LLMs on existing controlled generation datasets. This paper also introduce a numerical planning task. Experimental results show that LLMs struggle at fine-grained hard constraints.

**Reasons To Accept:**

1. This paper is well-organized and easy to follow.
1. This motivation is reasonable to me and has not been systematically studied before.
2. This paper conduct extensive experiments on a wide range of control generation tasks.

**Reasons To Reject:**

1.While this paper shows many findings, few of them are new to the community.
2.There should be more discussions about why LLMs struggle at fine-grained hard constraints and how to address these problems.
3.It would be better to include vicuna and falcon in Table-2, Table-3, and Table-5.

**Reproducibility:**

4: Could mostly reproduce the results, but there may be some variation because of sample variance or minor variations in their interpretation of the protocol or method.

**Reviewer Confidence:**

4: Quite sure. I tried to check the important points carefully. It's unlikely, though conceivable, that I missed something that should affect my ratings.

---

> ### Author Rebuttal · Authors · 2023-08-29
>
> We sincerely appreciate your feedback and would like to clarify some concerns as follows:
>
> **Reason 1: Few of findings are new to the community.**
>
> Many findings seem obvious in retrospect, but this does not mean that the community is already aware of them and can use them as building blocks for future work (https://aclrollingreview.org/reviewertutorial#6-check-for-lazy-thinking). To the best of our knowledge, we propose the numerical planning task and are the first to systematically study the performance of large language models on extensive controlled generation tasks, which have never been explored before and as such are new.
>
> **Reason 2: Discussion about why LLMs struggle at fine-grained hard constraints and how to address.**
> Thank you for the insightful comments. Our work focuses on analyzing the capabilities of large language models. We hesitate about sharing our speculation without knowing the details of models such as ChatGPT. For example, we speculate that the model does not perform on controlled paraphrase generation tasks due to the lack of linguistic knowledge in the pertaining. Including parsing corpus such as TreeBank in the pretraining may help boost the model performance. However, this is unverifiable. Therefore, we emphasize our contribution on the  extensive and thorough experimental design and showcase what we find out instead of risking to share hypotheses that we are not sure about.  We will have some discussion with Reviewer 2 around if tokenization would be the only reason for the poor performance on the NPB task, we invite you to take a look at our response below (Q. D).
>
>
> **Reason 3: Include Vicuna and Falcon in Table 2/3/5**
>
> Thank you for your suggestion! We conduct additional experiments to include Vicuna and Falcon throughout different tasks to make our study more comprehensive. Constrained by the computing power, we use Vicuna-7B for Vicuna and Falcon-7B-Instruct for Falcon. Unsurprisingly, we find that both models underperform ChatGPT. Thus, additional results do not affect our conclusion in the paper.
>
> Please see complementary results here, we will merge them with current results for the final version:
>
> Table 2 for Numerical Planning Benchmark (NPB):
>
> |     Model    | SE-Count | SR-Last Word | SR-both | MSE-count |
> |:------------:|:--------:|:------------:|---------|-----------|
> |  Vicuna_{zs} |   0.08   |     0.09     | 0.03    | 27.68     |
> | Vicuna_{ICL} |   0.13   |     0.30     | 0.04    | 13.43     |
> |  Falcon_{zs} |   0.13   |     0.42     | 0.08    | 11.60     |
> | Falcon_{ICL} |   0.11   |     0.34     | 0.03    | 13.72     |
>
> Table 3 for Content-Constrained Text Generation:
>
> |Model           | Topic  |   Sentiment  |     Keyword |
> | -------- | ------- |------- |------- |
> |Vicuna_{zs}  |	61.0	|           80.5	      |     91.6 |
> |Vicuna_{ICL} | 65.8	|	87.4	      |     94.3 |
> |Falcon_{zs} 	|61.9	|           81.0	      |	92.1 |
> |Vicuna_{ICL} 	| 66.0	|	87.7	      |	94.2 |
>
> Table 5 for Rationale Generation:
> | Model             |          Leakage      |    Non-Leakage |
> | -------- | ------- |------- |
> | I + R_{Vicuna} |		0.95	        |	 0.74   |
> | I + R_{Falcon} |		0.83	        |	 0.65   |

---

### Meta-Review · Area_Chair_qres · 2023-09-20

**Recommendation:** 3

**Metareview:**

This paper studies the controllability of large language models on generation tasks on various benchmark tasks. Reviewers found this paper well written and easy to follow. Reviewers also found the experiments extensive and well designed, covering a wide range of controlled generation tasks. While this is an analytic paper with comprehensive evaluations, it can add significant value by including an in-depth discussion on why LLM struggled at fine-grained hard constraints and how to close the gaps, as also commented by one reviewer. The authors provided some detailed response on this of which the author should consider add to the revised version of the paper.

---

### Decision · Program_Chairs · 2023-10-07

**Decision:**

Accept-Main

**Comment:**

This paper studies the controllability of large language models on generation tasks on various benchmark tasks. Reviewers found this paper well written and easy to follow. Reviewers also found the experiments extensive and well designed, covering a wide range of controlled generation tasks. While this is an analytic paper with comprehensive evaluations, it can add significant value by including an in-depth discussion on why LLM struggled at fine-grained hard constraints and how to close the gaps, as also commented by one reviewer. The authors provided some detailed response on this of which the author should consider add to the revised version of the paper.